# Scaling Language-Centric Omnimodal Representation Learning

**Chenghao Xiao**[1]    **Hou Pong Chan**[1,†]    **Hao Zhang**[1,†]
**Weiwen Xu**[1]    **Mahani Aljunied**[1]    **Yu Rong**[1,2,‡]
[1]DAMO Academy, Alibaba Group    [2]Hupan Lab

## Abstract

Recent multimodal embedding approaches leveraging multimodal large language models (MLLMs) fine-tuned with contrastive learning (CL) have shown promising results, yet the underlying reasons behind their superiority remain underexplored. This work argues that a crucial advantage of MLLM-based approaches stems from implicit cross-modal alignment achieved during generative pretraining, where the language decoder learns to exploit multimodal signals within a shared representation space for generating unimodal outputs. Through analysis of anisotropy and kernel similarity structure, we empirically confirm that latent alignment emerges within MLLM representations, allowing CL to serve as a lightweight refinement stage. Leveraging this insight, we propose a **L**anguage-**C**entric **O**mnimodal **Emb**edding framework, termed **LCO-EMB**. Extensive experiments across diverse backbones and benchmarks demonstrate its effectiveness, achieving state-of-the-art performance across modalities. Furthermore, we identify a **G**eneration-**R**epresentation **S**caling **L**aw (**GRSL**), showing that the representational capabilities gained through contrastive refinement scale positively with the MLLM's generative capabilities. This suggests that improving generative abilities evolves as an effective paradigm for enhancing representation quality. We provide a theoretical explanation of GRSL, which formally links the MLLM's generative quality to the upper bound on its representation performance, and validate it on a challenging, low-resource visual-document retrieval task, showing that continual generative pretraining before CL can further enhance the potential of a model's embedding capabilities.[1]

## 1   Introduction

Cross-modal representation alignment, such as vision-language alignment, has traditionally relied on massive-scale contrastive learning (CL) over paired cross-modal data, as seen in CLIP-style models [37, 50, 83]. Prior work primarily focuses on scaling model size, dataset volume, and batch size during training [9, 25, 50, 58, 83]. While these strategies demonstrate benefits in tasks like linear probing [9, 25, 50] and zero-shot classification [50, 83], performance tends to plateau on complex tasks requiring deeper cross-modal comprehension, *e.g.,* multilingual image retrieval [57, 60], visual text representations [11, 19, 74], and tasks involving interleaved multimodal encodings [69].

Recent approaches utilize autoregressive multimodal large language models (MLLMs) as the backbone models, followed by CL fine-tuning, to enhance representational capabilities, leading to improved performance on these complicated tasks [8, 39, 84]. However, the underlying reasons for the performance advantages of MLLM-based embedding approaches over traditional CLIP-based ones

---

[†]Corresponding authors, `kenchanhp@gmail.com`, `hzhang26@outlook.com`.
[‡]Project head.
[1]Codes, models, and resources are available at `https://github.com/LCO-Embedding/LCO-Embedding`.

39th Conference on Neural Information Processing Systems (NeurIPS 2025).

remain underexplored. This represents a critical research gap in understanding the limitations of CLIP-style models and the specific strengths MLLMs bring to these challenging scenarios.

To address this research gap, we conduct a systematic study of MLLM-based embedding models across modalities. First, we empirically investigate the embedding space patterns of MLLM representations before and after lightweight CL fine-tuning using only textual data, via anisotropy and kernel-level similarity. Our results show that *text-only* fine-tuning not only improves the discriminability of text embeddings but also generalizes to enhance the discriminability of embeddings in non-textual modalities. This finding reveals that *MLLMs achieve implicit cross-modal alignment during generative pretraining*, such that representation activation for one modality generalizes to others. We posit that the generative objective of MLLMs enables them to leverage multimodal information in the same semantic space by learning to generate textual outputs during pretraining. Thus, we argue that the knowledge foundation and intrinsic multimodal alignment established during generative pretraining grant MLLM-based embedding models the fundamental advantages.

Building on the observations, we propose a **L**anguage-**C**entric **O**mnimodal **Emb**edding framework, termed LCO-EMB, that employs language-centric paired data for efficient CL refinement. We highlight that *CL can function as a lightweight, post-hoc refinement step for mapping pre-aligned generative embeddings into a similarity-matching space* in MLLMs, which differs sharply from the computationally intensive CL required by CLIPs for alignment. Accordingly, this emerging paradigm shifts emphasis towards preserving the cross-modal alignment structure established during MLLM pretraining. In line with recent work [24, 86], LCO-EMB adopts LoRA [27] for representation activation of MLLM, aiming to enhance its representation capability with minimal disruption to the pretrained generative capabilities and latent multimodal alignment.

Extensive experiments across diverse backbones and benchmarks show that LCO-EMB outperforms state-of-the-art multimodal embedding models trained with much larger multimodal training sets, with text-only training sets. Combining minimal additional multimodal paired data in diverse formats further calibrates the embedding space of LCO-EMB for downstream tasks, setting a new state-of-the-art on MIEB [76], while also providing competitive performance on audio and videos. Further analysis reveals that LoRA with language-centric contrastive learning yields superior results compared to alternative fine-tuning strategies, suggesting the importance of preserving the latent alignment structure during CL through minimal modification to the MLLM's pretrained knowledge. CL acts less as a means of introducing new knowledge and more as a lightweight activation mechanism, serving primarily to project the embedding space into a similarity-matching subspace.

As LCO-EMB relies on the inherent multimodal alignment capability of MLLMs, we further investigate the relationship between potentials of representation quality and the underlying generative ability of MLLMs. Through experiments with backbones of various sizes and generation strengths, we identify a **Generation-Representation Scaling Law** (**GRSL**), indicating that multimodal representational capabilities gained through contrastive refinement scales positively with the MLLM's generative capabilities before CL. GRSL suggests that improving the MLLM's initial generative capability—via continued generative pretraining or supervised fine-tuning—is an effective strategy for enhancing its potential in multimodal representations. We offer a theoretical explanation for GRSL through a PAC-Bayesian generalization bound, showing that an MLLM's generative capability determines an upper bound for its representational potential. To empirically validate this, we introduce **SeaDoc**, the most difficult visual document retrieval task to date in low-resource Southeast Asian languages. Through continual OCR-intense pretraining in low-resource languages, we show that retrieval performance enhances after the same amount of text-only contrastive learning.

Our contributions are threefold: (1) We propose a language-centric omnimodal representation learning framework, achieving promising performance across various MLLM backbones and embedding benchmarks. (2) We identify a Generation-Representation Scaling Law (GRSL), that representational capabilities after CL scales positively with the MLLM's generative capabilities. (3) We provide a theoretical justification for GRSL, followed by comprehensive empirical studies, demonstrating that generative capability sets a fundamental upper bound on representational quality in MLLMs.

## 2 Latent Cross-Modal Alignment in MLLMs

In this section, we conduct an in-depth empirical analysis of multimodal large language models (MLLMs) to investigate *whether their internal representations exhibit latent cross-modal alignment*

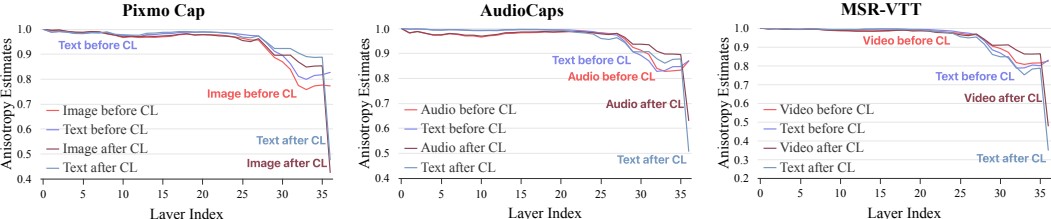

Figure 1: The anisotropy estimates of Qwen2.5-Omni-3B embeddings across text, image, audio, and video modalities. The vanilla model exhibits typical representation degeneration (anisotropy) for all modalities. After applying **text-only** contrastive learning, embeddings across modalities become more isotropic, indicating latent **language-centric** cross-modal alignment within the model.

through two geometric properties, *i.e.,* degree of anisotropy [23] and kernel-level similarity [29]. Specifically, starting with an MLLM, we directly take out its text decoder [30], *i.e.,* the LLM, and fine-tune it using text-only contrastive learning with LoRA on anchor-entailment text pairs from NLI datasets. Then, we merge the trained LoRA weights into the LLM and re-plug it into the original MLLM architecture. The detailed experimental settings are summarized in Section 4.1.

## 2.1 Analysis of Anisotropy Degrees

Language models trained on self-supervised objectives are known to suffer from anisotropy [17, 21], an embedding degeneration issue characterized by hidden representations collapsing into a confined region of representation space, resulting in high expected cosine similarity between random inputs. Contrastive learning is known to have the uniformity promise [68, 73] through enhancing discriminability across random negative pairs. Here, we employ contrastive learning to fine-tune multimodal large language models (MLLMs) exclusively with paired text data. We then compare the behaviors of models before and after fine-tuning to assess whether text-only training can effectively mitigate anisotropy for non-textual modalities, even in the absence of explicit multimodal training. The successful transfer of improvements across modalities would provide empirical evidence that MLLMs inherently preserve geometrically aligned latent spaces among different modalities.

We follow Ethayarajh [17] and Xiao et al. [73] to approximate the degree of anisotropy using the expected mean of cosine similarity between random data points. Let $\mathbf{h}_i, \mathbf{h}_j \sim \mathcal{D}$ be the embedding vectors sampled independently and identically distributed (*i.i.d.*) from the empirical distribution $\mathcal{D}$ of the representation space. Then, the degree of anisotropy is calculated as:

$$\text{Anisotropy} := \mathbb{E}_{\mathbf{h}_i, \mathbf{h}_j \sim \mathcal{D}} \left[ \cos(\theta_{ij}) \right] = \mathbb{E}_{\mathbf{h}_i, \mathbf{h}_j \sim \mathcal{D}} \left[ \frac{\mathbf{h}_i^\top \mathbf{h}_j}{\|\mathbf{h}_i\| \, \|\mathbf{h}_j\|} \right]. \tag{1}$$

In practice, we approximate it empirically using a finite sample of $N$ embeddings $\{\mathbf{h}_1, \ldots, \mathbf{h}_N\}$ as:

$$\hat{\mathbb{E}} \left[ \cos(\theta) \right] = \frac{2}{N(N-1)} \sum_{1 \le i < j \le N} \frac{\mathbf{h}_i^\top \mathbf{h}_j}{\|\mathbf{h}_i\| \, \|\mathbf{h}_j\|}. \tag{2}$$

Specifically, we use Qwen2.5-Omni-3B [77] as the backbone model and fine-tune it with text-only contrastive learning. To ensure objective and fair semantic comparison between text and other modalities, we utilize paired datasets, *i.e.*, Pixmo Cap [13] for image-text, AudioCaps [32] for audio-text, and MSR-VTT [79] for video-text, for anisotropy comparison. The changes in the embedding spaces of different modalities after the text-only contrastive learning are depicted in Figure 1. As anticipated, the embedding space produced by Qwen2.5-Omni-3B initially exhibits a collapsed structure and poorly separated distribution across modalities. After text-only contrastive learning, embedding spaces of non-text modalities surprisingly generalize to become *more isotropic, dispersing more uniformly across the respective subspaces*. **The generalized reduction in anisotropy for image, audio, and video embeddings reflects an underlying latent semantic alignment with textual representations within the base model.**

## 2.2 Analysis of Kernel-level Similarity

Building on the latent cross-modal alignment in MLLMs, we further employ kernel-level similarity to analyze the improvement in similarity structure alignment across modalities after fine-tuning.

Given a function $f : \mathcal{X} \to \mathbb{R}^n$ that maps inputs to high-dimensional representations, the associated kernel $K : \mathcal{X} \times \mathcal{X} \to \mathbb{R}$ characterizes the induced similarity structure via inner product $K(x_i, x_j) = \langle f(x_i), f(x_j) \rangle$, where $x_i, x_j \in \mathcal{X}$ and $K \in \mathcal{K}$. Then, a kernel alignment metric $m : \mathcal{K} \times \mathcal{K} \to \mathbb{R}$ is adopted to quantify the similarity between two kernels, *i.e.,* the "similarity of similarity structures", by assessing *how closely the distance metric induced by one representation space aligns with that of another*. Prior work [29] examines these structures across independently trained models and finds convergence in their representations. For instance, despite being trained separately, LLaMA [64] and DINOv2 [49] exhibit comparable similarity perception of captions and images from paired datasets.

Similar to Huh et al. [29], we adopt mutual $k$NN to quantify the overlap in the top-$k$ nearest neighbors of each data point shared across the similarity structures induced by two models, $f$ and $g$. The data samples $(x_i, y_i)$ are drawn in mini-batches of size $b$ from a distribution $\mathcal{X}$. Taking the image-text alignment as an example, each $(x_i, y_i)$ pair, *i.e.*, an image and its corresponding caption, is assumed to share the same semantic content, denoted as $x_i \triangleq y_i$. These paired samples serve as semantic anchors for evaluating the representations across modalities. Given the models $f$ and $g$,[2] the corresponding embeddings of each paired sample are attained as $\phi_i = f(x_i)$ and $\psi_i = g(y_i)$. For a mini-batch of $b$ data samples, we can derive the feature sets $\Phi = \{\phi_1, \dots, \phi_b\}$ and $\Psi = \{\psi_1, \dots, \psi_b\}$. For each feature $\phi_i \in \Phi$ (and similarly $\psi_i \in \Psi$), the $k$NN set $\mathcal{S}(\phi_i)$ (or $\mathcal{S}(\psi_i)$) comprises the indices of its $k$ nearest neighbors within its feature collection, excluding itself, which is determined by $d_{\text{knn}}$ as:

$$\mathcal{S}(\phi_i) = d_{\text{knn}}(\phi_i, \Phi \setminus \{\phi_i\}), \qquad \mathcal{S}(\psi_i) = d_{\text{knn}}(\psi_i, \Psi \setminus \{\psi_i\}). \tag{3}$$

The kernel-level similarity score $m_{\text{NN}}$ for a specific feature pair $(\phi_i, \psi_i)$ is the normalized cardinality of the intersection of their $k$NN sets:

$$m_{\text{NN}}(\phi_i, \psi_i) = \frac{1}{k} |\mathcal{S}(\phi_i) \cap \mathcal{S}(\psi_i)|, \tag{4}$$

which indicates the proportion of shared nearest neighbors, where $k$ denotes the number of nearest neighbors. The overall $m_{\text{NN}}$ metric is computed as the average of the individual $m_{\text{NN}}(\phi_i, \psi_i)$ scores across the mini-batch.

Different from Huh et al. [29], we utilize kernel alignment metrics to inspect cross-modal alignment within the same model. Specifically, we attain hidden representations at all layers from the 3B and 7B variants of Qwen2.5-VL-Instruct, and assess the self-similarity between the vision and language kernels using Equation 4. The text-only contrastive learning is applied to the language decoder, and alignment scores are compared before and after fine-tuning. As illustrated in Figure 2, two notable findings emerge: (1) cross-modal kernel alignment improves after text-only contrastive learning, indicating the presence of inherent latent alignment across modalities; and (2) the 7B variant exhibits consistently stronger cross-modal kernel alignment than the 3B variant, both before and after fine-tuning. This advantage may be attributed to the expanded parameter space of the larger model, yielding better expressivity and a superior ability to capture latent cross-modal

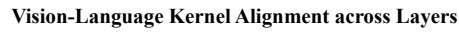
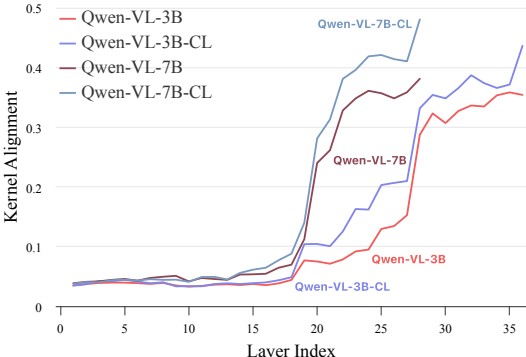

Figure 2: Layer-wise vision-language kernel alignment before and after text-only contrastive learning, evaluated on Qwen-VL models with 7B (28 layers) and 3B (36 layers) parameters. Note the 3B model has more layers than the 7B model.

relationships during pre-training. Collectively, these findings suggest that **inherent cross-modal binding enables the optimization of representation in one modality to induce corresponding improvements in other modalities**.

## 3    Language-centric Omnimodal Representation Learning

The preliminary experiments reveal that MLLMs implicitly acquire cross-modal alignment during pretraining. Although initial embeddings are suboptimal for similarity matching, latent align-

---

[2]In this example, we use the same model to encode the image $x_i$ and its caption $y_i$, *i.e.*, $f = g$.

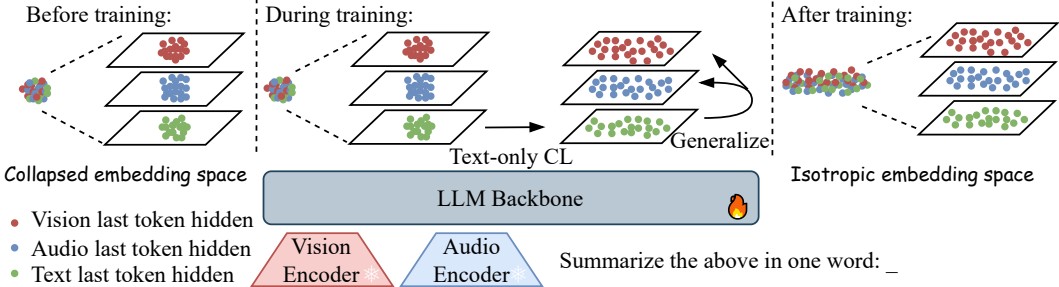

Figure 3: The power of language-centric omnimodal representation learning: Before text-only contrastive learning (CL), representations across modalities in multimodal large language models (MLLMs) exhibit anisotropy, collapsing into a confined subspace. Text-only CL disperses textual representations by increasing their separation, effectively reducing anisotropy. Notably, this process generalizes to alleviate anisotropy in non-textual modalities, despite the absence of direct supervision.

ment emerges in intermediate layers. This inherent alignment can be efficiently unlocked through lightweight text-only contrastive fine-tuning, enhancing representation quality across both textual and non-textual modalities. Building on this insight, we introduce **L**anguage-**C**entric **O**mnimodal representation learning (LCO-EMB), a framework that leverages language-centric data and lightweight contrastive learning to boost MLLM representation capabilities across modalities.

The contemporary architectures of MLLM are composed of modality-specific encoders, a projector, and a language decoder (*i.e.,* an LLM), with the projector aligning modality-specific representations to the decoder's embedding space [2, 40, 59, 77]. For text-only variants of LCO-EMB, we *isolate and fine-tune only the language decoder via text-only contrastive learning*, while freezing the parameters of modality encoders and the projector. After training, the updated decoder is re-plugged into the original model. We further incorporate minimal multimodal paired data to calibrate the embedding space for downstream tasks, resulting in multimodal variants of LCO-EMB.

Central to our method is the preservation of the latent cross-modal alignment established during generative pretraining. This alignment, wherein multimodal embeddings are integrated into a shared latent subspace by the language decoder, is fundamental to the model's multimodal representation capability. We employ LoRA [27], which introduces low-rank trainable parameters into select layers while freezing the original model. While LoRA is widely recognized for enabling parameter-efficient fine-tuning, its primary advantage in our context is its ability to minimally perturb the original model. This approach yields two critical benefits: (1) *it preserves the model's generative capabilities through minimal weight modifications* [4]; and (2) *it maintains the latent cross-modal alignment, especially in the language decoder's embedding layer, which is unaffected by the adaptation*.

## 4 Experiments

### 4.1 Experimental Settings

**Backbones and hyperparameters.** We use LLaVA-Next [41], Qwen2.5-VL [2], and Qwen2.5-Omni [77] as backbones, all conforming to the standard architecture of modality-specific encoders, a projector, and a language decoder. LLaVA-Next and Qwen2.5-VL focus on image/video-text, while Qwen2.5-Omni supports omnimodal inputs, covering text, image, video, and audio. We utilize the 8B variant of LLaVA-Next, the 3B and 7B variants for both Qwen2.5-VL and Qwen2.5-Omni. We adopt AdamW optimizer with a cosine learning rate schedule, a peak learning rate of $4 \times 10^{-4}$, and a batch size of $768$,[3] to train the model for 2 epochs. The LoRA rank ($r$) and $\alpha$ are set as $64$ and $16$ for text-only variants and $64$ and $128$ for multimodal variants, respectively. For multimodal variants of Qwen2.5-Omni-7B, we use a reduced learning rate of $3 \times 10^{-4}$ due to the loss spike.

**Datasets.** *(1) Text-only Setting.* We consider two data settings: **all-NLI** and **Scale-1M**. The **all-NLI** combines MNLI [70] and SNLI [5], both frequently used for sentence representation learning. Each

---

[3]For multimodal variants with additional image-text and interleaved data, we scale batch size by the ratio of total to text-only data size. Thus, for our 370k data (276k text-only), the batch size is $1,052$ ( $1.37 \times 768$).

Table 1: **MIEB-Lite (51 tasks) results broken down by task categories.** We provide averages of both English and multilingual tasks. Models are ranked by the Mean (m) column. Shortcuts are x="Crosslingual", m="Multilingual", en="English", and task categories from MIEB [76]. We refer to the latest MIEB leaderboard to obtain scores for the compared baselines.

| Model Name (↓) | Dataset Size | Rtrv. (11) | Clus. (2) | ZS. (7) | LP. (8) | Cmp. (6) | VC. (5) | Doc. (6) | vSTS (en) (2) | Rtrv. (m) (2 (44)) | vSTS (x&m) (2 (19)) | Mean (en) (47) | Mean (m) (51) |
|---|---|---|---|---|---|---|---|---|---|---|---|---|---|
| **Encoder baselines** | | | | | | | | | | | | | |
| CLIP-ViT-bigG [34] | 2B | 34.2 | 80.8 | 72.4 | 77.8 | 35.0 | 43.0 | 35.5 | 73.4 | 26.2 | 34.5 | 56.5 | 51.3 |
| SigLIP-so400m [83] | 9B | 32.4 | 75.9 | 73.8 | 78.8 | 32.8 | 48.0 | 46.9 | 69.6 | 35.4 | 41.4 | 57.3 | 53.5 |
| **MLLM baselines** | | | | | | | | | | | | | |
| VLM2Vec-LoRA [31] | 662k | 21.0 | 66.4 | 32.1 | 64.8 | 29.4 | 65.3 | 42.7 | 70.9 | 24.8 | 42.2 | 49.1 | 46.0 |
| E5-V [30] | 276k | 26.9 | 51.7 | 36.2 | 70.6 | 39.4 | 52.6 | 56.0 | 81.2 | 58.3 | 46.3 | 51.8 | 51.9 |
| Voyage Multimodal 3 [65] | - | 33.2 | 76.6 | 48.6 | 69.3 | 35.8 | 50.0 | 63.5 | 84.2 | 49.0 | 70.4 | 57.7 | 58.1 |
| mmE5 (11B) [8] | 2.1M | 34.2 | 77.0 | 59.8 | 71.1 | 27.8 | 59.2 | 53.9 | 78.8 | 66.6 | 54.6 | 57.7 | 61.8 |
| GME (7B) [84] | 8.0M | 37.9 | 69.6 | 55.5 | 68.7 | 52.2 | 55.4 | 86.1 | 81.8 | 62.4 | 75.4 | 63.4 | 64.5 |
| **Our *Text-only Variants*** | | | | | | | | | | | | | |
| LCO-Emb-VL (3B) | 276k | 32.6 | 61.8 | 45.0 | 67.4 | 38.5 | 57.7 | 62.2 | 86.1 | 52.4 | 76.5 | 56.4 | 58.0 |
| LCO-Emb-VL (7B) | 276k | 31.8 | 52.7 | 49.1 | 68.5 | 40.4 | 63.1 | 66.0 | 88.4 | 59.8 | 84.3 | 57.5 | 60.4 |
| **Our *Multimodal Variants*** | | | | | | | | | | | | | |
| LCO-Emb-VL (3B) | 370k | 34.0 | 71.6 | 58.1 | 68.3 | 46.1 | 57.8 | 73.0 | 83.8 | 54.6 | 76.1 | 61.6 | 62.3 |
| LCO-Emb-VL (7B) | 370k | 36.4 | 76.0 | 66.8 | 72.5 | 51.0 | 65.2 | 75.6 | 86.6 | 63.1 | 83.3 | _66.2_ | _67.6_ |
| LCO-Emb-Omni (3B) | 370k | 34.5 | 78.3 | 66.1 | 72.8 | 48.2 | 59.2 | 73.4 | 85.7 | 54.5 | 80.4 | 64.8 | 65.3 |
| LCO-Emb-Omni (7B) | 370k | 36.4 | 80.0 | 68.5 | 74.1 | 50.1 | 70.5 | 75.4 | 86.2 | 64.3 | 82.4 | **67.6** | **68.8** |

instance includes a premise with three hypotheses (entailment, neutral, contradiction). We use ∼276k triplets from all-NLI with entailments as positives and contradictions as hard negatives. We further construct **Scale-1M**, a curated collection of 1M sentence pairs sampled from 20M multilingual parallel corpora, including Global Voice [48], MUSE [53], News Commentary [62], Tatoeba [1], Talks [52], WikiMatrix [55], and other Sentence Transformers sources [51]. This design leverages diverse descriptive text to simulate image captions, activating image representations without direct image supervision, and integrates multilingual pairs to enhance cross-lingual alignment, which may in turn enhance multimodal alignment across languages. *(2) Multimodal Setting.* Building on all-NLI, we further add ∼94k synthetic multimodal pairs (Appendix B) to enhance alignment in the downstream task format space, yielding a final dataset of ∼370k triplets.

**Benchmarks.** For *image-text embedding tasks*, we adopt **MIEB-Lite** (51 tasks), the official lightweight version of MIEB (130 tasks; [76]), which covers eight categories (Appendix C), including Linear Probing, Retrieval (English and Multilingual), Zero-shot Classification, Compositionality Evaluation, Vision-centric QA, Document Understanding, Clustering, and Visual STS (English and Cross-lingual). For rapid iteration and ablation, we employ a compact subset of 18 overlapping tasks, termed **MIEB-Sub18** (Appendix D). For *audio-text embedding tasks*, we evaluate on AudioCaps [32] and Clotho [16] datasets. For *video-text embedding tasks*, we utilize MSR-VTT [79] and ActivityNet [26] datasets. The performance on these tasks provides complementary evidence supporting the universality and effectiveness of LCO-Emb, extending beyond the vision and language modalities. For both audio-text and video-text embedding tasks, we utilize the *Recall@1* as the evaluation metric.

## 4.2 Performance Comparison on the MIEB Benchmark

To better understand the results, we brief the goal and metric of each MIEB-Lite category: (1) **Visual STS** reformulates semantic textual similarity as a vision task by rendering text as images to test models' semantic understanding, evaluated by *Spearman correlation*; (2) **Document Understanding/Visual Document Retrieval** measures a model's ability to capture layout-aware textual semantics in visual documents and image-text alignment, evaluated by *nDCG@5*; (3) **Image Linear Probing** assesses the discriminative and transferable quality of frozen visual representations using *accuracy*; (4) **Compositionality Evaluation** tests fine-grained image-text alignment with *accuracy*; (5) **Vision-centric QA** evaluates visual reasoning and understanding through *accuracy*; (6) **Retrieval** measures modality-specific and joint encoding performance with *nDCG@10*; (7) **Zero-shot Classification** evaluates similarity-based classification using *accuracy*; and (8) **Clustering** examines the structural coherence of embeddings using the *NMI* metric. Detailed task descriptions are in Appendix C.

We evaluate LCO-Emb on the 51 tasks of the MIEB-Lite benchmark. As reported in Table 1, LCO-Emb consistently outperforms strong baselines, including E5-V [30], VLM2Vec [31], Voyage-

Table 2: Performance and efficiency comparisons of different training strategies using 3B and 7B variants of Qwen2.5-VL backbones. GPU hours are benchmarked by hours × number of H20 GPUs.

| Training Strategy | Training Time (GPU Hours) | Multiling. Img. Rtr | V-STS (Eng.) | V-STS (cross) | V-STS (multi) | Doc. Und. | Linear Probe | Average |
|---|---|---|---|---|---|---|---|---|
| Qwen2.5-VL-3B | n/a | 31.73 | 73.82 | 59.03 | 68.57 | 28.82 | 46.96 | 51.49 |
| w/ CLIP-style CL (multimodal) | ~453.0 Hours | 25.15 | 72.51 | 67.45 | 65.22 | 48.91 | 41.05 | 53.38 |
| w/ Linear Proj. (text-only) | ~4.5 Hours | 31.31 | 75.25 | 62.95 | 69.32 | 28.12 | 49.19 | 52.69 |
| w/ Full-Finetune (text-only) | ~8.5 Hours | 44.61 | 81.65 | 68.67 | 77.75 | 49.71 | 50.21 | 62.10 |
| w/ LoRA (text-only) | ~4.7 Hours | **51.61** | **81.88** | **74.97** | **78.30** | **57.90** | **53.05** | **66.28** |
| Qwen2.5-VL-7B | n/a | 40.31 | 73.82 | 59.03 | 68.56 | 28.82 | 46.96 | 52.92 |
| w/ CLIP-style CL (multimodal) | ~550.0 Hours | 18.24 | 73.92 | 68.70 | 65.41 | 44.89 | 38.93 | 50.02 |
| w/ Linear Proj. (text-only) | ~8.8 Hours | 40.29 | 72.05 | 65.46 | 70.88 | 35.69 | 52.96 | 56.22 |
| w/ Full-Finetune (text-only) | ~17.3 Hours | 44.05 | 83.15 | 79.09 | 81.28 | 58.02 | 53.34 | 66.49 |
| w/ LoRA (text-only) | ~9.3 Hours | **56.64** | **85.05** | **85.30** | **83.48** | **67.49** | **53.91** | **71.98** |

Multimodal-3 [65], mmE5 [8], and GME [84]. Remarkably, despite using only ~0.37M training pairs—about 21× less data than GME (~8M)—our multimodal variants set a new state-of-the-art on MIEB. Consistent with findings from Xiao et al. [76], MLLM-based embedding models excel at tasks leveraging MLLM backbones' reasoning and cross-modal understanding abilities, such as multilingual alignment, compositionality, and document understanding. Beyond these strengths, LCO-EMB also attains competitive results on clustering, linear probing, and zero-shot classification—areas where MLLM-based representations typically lag behind CLIP-style models. Notably, even our text-only variants, trained with minimal text-only contrastive data, surpass advanced proprietary model Voyage-Multimodal-3. Incorporating only ~94k additional multimodal samples (image–text and interleaved data; Appendix B) further calibrates the representation space for downstream task formats, resulting in a compact yet highly effective dataset of ~370k triplets.

### 4.3 Analysis of Representational Capability of LCO-EMB

To better analyze the representational capability of LCO-EMB, we use the text-only variants, *i.e.*, without utilizing the synthetic multimodal pairs, as evaluation targets and conduct extensive validation and ablation studies on **MIEB-Sub18** benchmark.

**Main results.** We assess text-only variants of LCO-EMB against the advanced embedding methods on the MIEB-Sub18. As illustrated in Figure 6 (Appendix D), LCO-EMB, trained on the 3B and 7B variants of Qwen2.5-VL-Instruct (VL) and Qwen2.5-Omni (Omni), consistently outperforms the leading embedding models across a variety of multimodal downstream tasks. On average across all evaluation categories, the text-only variants of LCO-EMB have outperformed E5-V [30] and Voyage-Multimodal-3 [65] by 21.69 and 13.00 points, where E5-V and Voyage-M3 are advanced open-source and proprietary MLLM embedding models, respectively. Notably, LCO-EMB delivers significant improvements on the Linear Probing, Cross-lingual Visual STS, and Multilingual Image Retrieval tasks, outperforming prior advanced methods by margins of 21.02, 10.26, and 15.35 points, respectively. The results highlight the effectiveness and generalizability of LCO-EMB. It is also noteworthy that while Voyage-M3 is a commercial model explicitly optimized on PDF–text pairs for document understanding tasks, LCO-EMB, trained solely on textual data, still achieve comparable results.

**Comparison of different training strategies.** We apply LoRA to enhance representational capacity while preserving latent cross-modal alignment. To assess this design, we experiment with Qwen2.5-VL 3B and 7B backbones, comparing LoRA against three baselines: (1) *standard CLIP-style contrastive fine-tuning* on 800K PixmoCaps image-caption pairs, (2) *full fine-tuning*, and (3) a *shallow projection* that adds a linear layer after the output. Reported in Table 2, the CLIP-style baseline underperforms text-only LoRA, requires 50× more training time, and the shallow projection increases parameters but does not effectively leverage pretrained cross-modal structure, yielding only marginal gains over native embeddings. Full fine-tuning achieves reasonable results but remains notably inferior to LoRA. We attribute this gap to an objective mismatch: contrastive loss deviates from the model's pretraining objective, and full fine-tuning consequently induces larger perturbations to the pretrained parameters, which are more likely to disrupt the established cross-modal alignment. Detailed analysis of LoRA hyperparameters[4] is presented in Appendix E. We further analyze the

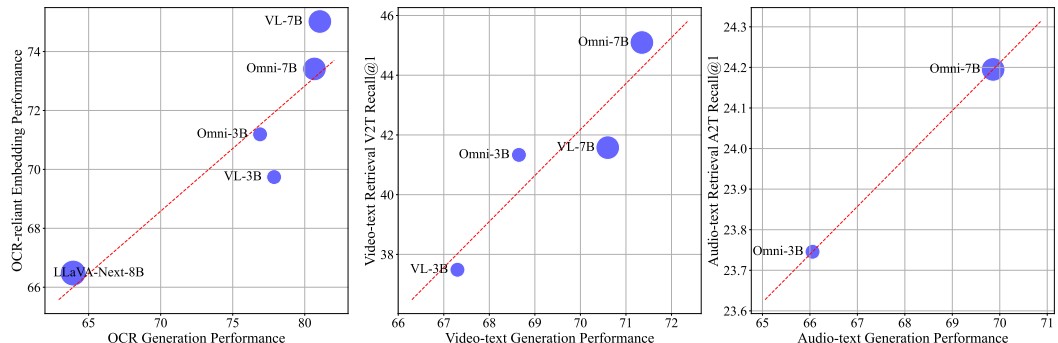

Figure 4: Scaling relationship between generation benchmark performance (X-axis) and representation benchmark performance after language-centric contrastive learning (Y-axis).

impact of different training datasets and model merging technique in Appendix F.

## 5 Generation-Representation Scaling Law

The superior performance of LCO-EMB is primarily attributed to the intrinsic multimodal alignment capabilities of the backbone MLLMs, which we activate through lightweight contrastive fine-tuning. This observation prompts a fundamental question: *What is the relationship between the inherent multimodal generative ability of MLLMs and their representation potential?* Through empirical analysis, we reveal a positive scaling correlation between these two aspects. Furthermore, we substantiate our empirical findings with a theoretical analysis with a PAC-Bayesian generalization bound, linking models' generative capabilities with the upper bound of their representation performance.

### 5.1 The Relationship between Generative and Representational Capabilities

We conduct an empirical analysis to investigate the relationship between improving multimodal representation capabilities via text-only contrastive learning and the intrinsic generative capacity of MLLMs. Our analysis spans three different types of modality pairs, *i.e.,* OCR-based image-text tasks, video-text tasks, and audio-text tasks. The experimental setup is detailed below:

- **OCR-Based Image-Text Tasks:** We evaluate OCR-dependent capabilities through paired representation and generative tasks. For representation tasks, we average scores from Visual Semantic Textual Similarity (V-STS-English) and Document Understanding. Generative performance is measured by averaging results from TextVQA [56], DocVQA [45], OCRBench [42], and ChartQA [44].
- **Video-Text Tasks:** Representation capabilities are assessed using video-text Recall@1 scores averaged across MSR-VTT [78] and ActivityNet [6]. Generative performance combines results from Video-MME$_{w/ sub}$ [20] and MVBench [38].
- **Audio-Text Tasks:** We compute Recall@1 scores using Clotho [16] and AudioCaps [32]. For generative evaluation, we average performance on MMAU [54] and VoiceBench [10], comprehensive benchmarks encompassing multiple sub-tasks.

**Results.** As depicted in Figure 4, we observe a consistently positive correlation between baseline generative performance before CL and the post-CL representational performance across different MLLM backbones on all task categories. This observation leads to the discovery of **G**eneration-**R**epresentation **S**caling **L**aw (**GRSL**), where *the representational abilities of MLLMs, enhanced through contrastive refinement, scale positively with the model's original generative capability*. This insight suggests an alternative pathway for advancing multimodal models by harnessing the scaling effects of generative capacity. Next, we provide a theoretical analysis of GRSL, which formally links the MLLM's generative quality to the upper bound on its final embedding performance.

---

[4]There is a slight difference in the evaluation resolution between Table 4 and Table 2 due to different codebase versions used for the experiments.

## 5.2 Theoretical Analysis of Generation-Representation Scaling Law

We aim to prove that a stronger generative prior of MLLMs leads to better representations after contrastive fine-tuning. We formalize this intuition using the PAC-Bayesian framework. We begin by defining our central hypothesis, deferring formal definitions of population/empirical risk ($\mathcal{L}_c^{\text{pop}}, \hat{\mathcal{L}}_c^{\text{emp}}$) and the generative quality of the prior ($I_P(X;Y)$) to Appendix G.

### 5.2.1 Central Hypothesis

The core of our argument is that a good generative prior provides a "warm start" for contrastive fine-tuning. We formalize this as follows.

**Hypothesis 1** (Generative Warm Start). *Let $P$ be a generative prior and $Q$ the posterior after contrastive fine-tuning. The expected empirical loss under $Q$ is bounded by:*

$$\mathbb{E}_{\theta \sim Q} \left[ \hat{\mathcal{L}}_c^{\text{emp}}(\theta) \right] \leq \log N - I_P(X;Y) + \epsilon_P, \tag{1}$$

*where $\epsilon_P \geq 0$ captures the gap between the information-theoretic optimum and the loss achieved after finite-step contrastive fine-tuning. A better prior (higher $I_P$) leads to a smaller $\epsilon_P$.*

A detailed justification for this hypothesis, linking $I_P(X;Y)$ to the pre-existing alignment of representations, is provided in Appendix G.2.

### 5.2.2 Main Theoretical Result

**Theorem 1** (Generative-Contrastive PAC-Bayes Bound). *Let $P$ be a generative prior and $Q$ the posterior after contrastive fine-tuning on $n$ samples. Under Hypothesis 1, with probability at least $1 - \delta$, the expected population contrastive risk is bounded by:*

$$\mathbb{E}_{\theta \sim Q} \left[ \mathcal{L}_c^{\text{pop}}(\theta) \right] \leq \underbrace{\log N - I_P(X;Y)}_{\text{Generative Bottleneck}} + \underbrace{\epsilon_P}_{\text{Inefficiency Gap}} + \underbrace{\sqrt{\frac{\text{KL}(Q\|P) + \log(1/\delta)}{2n}}}_{\text{PAC-Bayes Complexity Penalty}}. \tag{2}$$

The proof is provided in Appendix H.

**Corollary 1** (Generative Performance Governs Representation Bound). *Using the approximation $I_P(X;Y) \approx H(Y) - \mathcal{L}_g(P)$ (see Appendix G), the bound is directly governed by the prior's generative loss:*

$$\mathbb{E}_{\theta \sim Q} \left[ \mathcal{L}_c^{\text{pop}}(\theta) \right] \lesssim \mathcal{L}_g(P) + (\log N - H(Y)) + \epsilon_P + \sqrt{\frac{\text{KL}(Q\|P) + \log(1/\delta)}{2n}}. \tag{3}$$

*This result formalizes our central claim: a **lower generative loss** $\mathcal{L}_g(P)$ directly tightens the upper bound on the final contrastive performance.*

**Interpretation of the Bound.** The theorem and its corollary reveal three distinct factors that govern the final representation quality:

1. **The Generative Bottleneck ($\log N - I_P(X;Y)$):** The final quality is limited by the mutual information, $I_P(X;Y)$, captured by the generative prior. A stronger generative model (higher $I_P$) lowers this performance floor.

2. **The Optimization Inefficiency ($\epsilon_P$):** A better prior creates a more favorable optimization landscape, resulting in a smaller "inefficiency gap" $\epsilon_P$.

3. **The Fine-tuning Cost ($\sqrt{\cdots}$):** The PAC-Bayes complexity penalty justifies using parameter-efficient methods (like LoRA) to keep $\text{KL}(Q\|P)$ small, thus retaining the benefits of the strong generative prior.

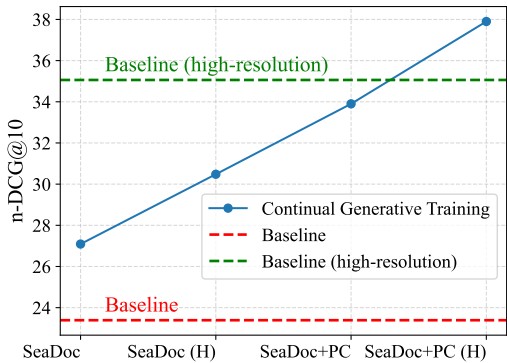

Figure 5: Retrieval performance of Qwen2.5-VL-3B fine-tuned on various continual generative fine-tuning strategies before CL on SeaDoc benchmark. PC denotes PixmoCaps. H denotes training with high resolution.

We further empirically investigate whether improving generative capabilities before contrastive learning raises the representational upper bound, which is the key aspect of GRSL. We construct a difficult Visual Document Retrieval (VDR) task: **SeaDoc**, a VDR task in low-resource **S**outh**E**ast **A**sian languages. Figure 5 summarizes the results for Qwen2.5-VL-3B with four continual pre-training settings: SeaDoc-train, SeaDoc-train + PixmoCaps (PC) to preserve general image perception capabilities, and training both settings at high resolution (H). The trend confirms that representation performance after CL improves as generative capability strengthens. Full details of SeaDoc construction, experimental settings, and analyses are provided in Appendix J.

## 6 Related Work

**Omnimodal Representation Learning.** Existing approaches to omnimodal representation learning [22, 67] typically rely on large-scale cross-modal pairs to train modality-specific encoders. Recent progress [8, 39, 84] highlights the potential of MLLMs for image–text alignment. However, the effectiveness of exploiting the latent alignment inherent in MLLMs' generative capabilities for omnimodal representation learning—and its underlying theoretical basis—remains unexplored.

**Modality-centric Representation Learning.** Prior work explores representation learning for a single modality. For instance, ImageBind [22] leverages the image modality as the anchor for contrastive learning to align with all other modalities. Web-SSL [18] explores language-free (thus "vision-centric") visual representation learning, which scales data volume to be on par with CLIPs to train DINOv2. By scaling up data volume, the vision-centric self-supervised learning can achieve OCR performance on par with CLIP, which is typically thought to attain through textual supervision [63]. E5-V [30] leverages text-only learning to generalize to images and composed retrieval tasks. We extensively study the language-centric view to train omnimodal representation models.

**Representation Capabilities.** Through investigating 50 models across 130 tasks in 39 languages, Xiao et al. [76] report that CLIP's performance gains from scaling data, batch size, and model size have largely plateaued on advanced representation benchmarks, including interleaved encodings [69, 85], compositionality [61], textual visual representations [19, 74], and image–multilingual text alignment [57]. They further highlight MLLM-based embedding models as a promising alternative, motivating our exploration of the relationship between representational and generative capabilities in MLLMs. Prior work has explored this connection: Cambrian-1 [63] combines a shared language decoder with various vision encoders for training MLLMs and demonstrates that the downstream performance of MLLMs scales with the representation capabilities of the vision encoders, while Yang et al. [80] formalizes the law between visual representation and MLLM generative capabilities. In contrast, we explore a fundamentally different concept: the "*Generation-Representation Scaling law*" between generation and representation capabilities of the MLLM itself. We see above as the "*Representation-Generation Scaling Law*" where the MLLM's generation performance scales with the strength of modality-specific encoders. In this work, we explore a fundamentally different concept: the "*Generation-Representation Scaling law*" where the MLLM's representation abilities scale with its own generation capabilities. Our findings align closely with Xiao et al. [75], who demonstrate that LLM-based embeddings excel at instruction following and reasoning-oriented retrieval.

## 7 Conclusion

In this work, we reveal that the superior performance of MLLM-based embedding approaches originates from implicit cross-modal alignment established during generative pretraining, wherein the language decoder learns to integrate multimodal information within a unified representation space. Leveraging this insight, we develop LCO-EMB, a language-centric omnimodal embedding framework that treats contrastive learning as a lightweight refinement stage, enhancing representation quality while preserving the model's generative capabilities. Building on this formulation, we introduce the Generation-Representation Scaling Law (GRSL), which establishes a positive correlation between a model's generative capacity and the representation upper bound. Our theoretical analysis, through a PAC-Bayesian generalization bound, together with extensive empirical validation on diverse and challenging benchmarks, confirms both the efficacy of LCO-EMB and the generality of GRSL. Collectively, these findings re-conceptualize the role of contrastive learning and position generative pretraining—not merely the expansion of cross-modal data—as the central driver of scalable, efficient, and robust multimodal representation learning. We discuss the limitations of this work in Appendix A.

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

# A  Limitations

In this work, we have studied the scaling law between generative capabilities of pretrained MLLMs, their latent multimodal alignment, and their representational capabilities after contrastive learning. We use MLLMs that have gone through generative pretraining and those that have attained different levels of generative capabilities, and let them go through lightweight contrastive learning. During contrastive learning, model weights are minimally adjusted, through low-rank adaptation, to project the original knowledge space into an embedding space suitable for similarity matching. However, we do note that one can also jointly train generative loss and contrastive loss [43, 46] to maintain a model's knowledge (through continual generative training), and enhance its representational power (through continual contrastive learning). Due to the high computational cost of this approach, we leave it as a promising direction for future work in the context of omnimodal representation learning.

# B  Details of Additional Multimodal Data

On top of the main text-only setting, the all-NLI training corpus—which plays a crucial role in unlocking the model's representational capacity—we further construct approximately 94k multimodal training samples to align the embedding space with the downstream multimodal task space. Specifically, we include: (1) **Visual Document.** Unlike most prior studies, we intentionally construct only about 23k triplets from Colpali [19] and Docmatix [35], rather than performing exhaustive data exposure. We found that large-scale visual document data, when not balanced with text and other task datasets, can degrade overall task generalization. (2) **Retrieval and Compositionality.** We include only 3k triplets from MS-COCO, aiming to introduce basic image–text alignment. To enhance robustness to varying input lengths, we apply augmentation techniques from LA(SER)$^3$ [72]. Interestingly, this not only improves length robustness but also enhances the model's spatial perception and image–text compositional reasoning. (3) **Multilingual/Diverse Text Data.** To enhance linguistic and contextual diversity, we sample several thousand examples from our Scale-1M dataset introduced in the main paper. (4) **General Synthetic Data.** We further construct around 60k synthetic samples in diverse formats to maintain and reinforce the model's instruction-following and interleaved alignment capabilities—which is important for tasks like VQA under the Reasoning-as-Retrieval paradigm. The diverse synthetic data also benefits classification tasks, improving both probing and zero-shot performance.

# C  Details of MIEB-Lite Benchmark

The MIEB-Lite benchmark comprises 51 tasks in 8 categories, where the details of each category are summarized as follows:

- **Visual STS** [74]: It conceptualizes traditional semantic textual similarity (STS) as a vision task by rendering text as images and evaluating the semantic understanding of visual encoders. Similarity scores are computed from the embeddings of image-text pairs and compared against human annotations using Spearman correlation. This task comprises three subcategories: *English* (STS 13 and STS 15), *cross-lingual* (STS-17, with image pairs in different languages, *e.g.,* Arabic–English), and *multilingual* (STS-b, with pairs in the same language, *e.g.,* Italian–Italian). Visual STS naturally assesses a model's interleaved encoding ability to capture semantic meaning from text in image form, with **Spearman correlation** as the primary evaluation metric.
- **Document Understanding/Visual Document Retrieval**: MIEB-lite selects 6 tasks from the Vidore benchmark [19], which is to retrieve visual documents that contain information to solve the problem in the query. This task assesses a model's ability to understand the complex layouts and textual information in visual documents, and the interleaved image-text alignment. Here we use **nDCG@5** as the evaluation metric.
- **Image Linear Probing**: MIEB-lite selects 8 challenging linear-probing datasets, including Country211, DTD, EuroSAT, GTSRB, OxfordPets, PatchCamelyon, RESISC45, and SUN397, which MLLMs typically struggle compared to CLIP-style models, as indicated by the MIEB benchmark leaderboard. We follow Xiao et al. [76] to adopt 16-shot linear probing, which closely preserves ranking compared to full-dataset probing, and report **accuracy** as the metric.
- **Compositionality Evaluation**: It evaluates fine-grained alignment of image-text features, requiring retrieving the groundtruth texts corresponding to the correct composition of all elements, *e.g.*, an

accurate fine-grained caption of an image, and vice versa for images given texts. This category includes ARO-Benchmark [82] and Winoground [61]. Here we use **accuracy** as the evaluation metric.

- **Vision-centric QA**: Given an interleaved input composed of a question conditioned on an image, the task requires models to retrieve the correct answer under the reasoning-as-retrieval paradigm [75]. This task category is mostly made of tasks assessing vision-centric capabilities [63], such as spatial relation perception, depth estimation, and relative distance. Here we use **accuracy** as the evaluation metric.

- **Retrieval**: MIEB-Lite adopts 11 retrieval tasks, consisting of image-only retrieval, image-text retrieval, and interleaved retrieval, providing a comprehensive assessment of models' modality-specific and composed encoding capabilities. In addition, it also selects WIT datasets [57] and XM3600 [60], totally covering image retrieval tasks across 38 different languages, *i.e.*, multilingual image retrieval, to assess a model's alignment capability between image and multilingual text embeddings, using **nDCG@10** as the primary metric.

- **Zero-shot Classification**: Zero-shot Classification assesses classification in a similarity-matching fashion. We use text prompts like "an image of a {label}" following common practices Radford et al. [50] and Xiao et al. [76]. MIEB-lite selects 7 challenging fine-grained zero-shot classification tasks, including CIFAR100, Country211, FER2013, FGVCAircraft, Food101, OxfordPets, and StanfordCars. Here we use **accuracy** as the evaluation metric.

- **Clustering**: Clustering provides an extra lens to inspect the clustered structure of embeddings. MIEB-lite adopts two clustering tasks, including fine-grained tasks such as Imagenet-Dog15 [14], which MLLM-based embedding models typically fail compared to CLIP-style models [76]. The **Normalized Mutual Information (NMI)** is utilized as the main evaluation metric.

# D  Details of MIEB-Sub18 Benchmark

We further select a smaller-scale subset than MIEB-lite, including 18 tasks from MIEB Xiao et al. [76] as **MIEB-Sub18**, which comprises 47 subtasks that are considered most challenging to the image-text embedding models, particularly in evaluating the capabilities of visual text representation, multilingual understanding, and interleaved encodings. Specifically, we focus on Visual STS [74], multilingual image retrieval [57], and document understanding from Vidore [19]. Additionally, we assess three image linear probing tasks where MLLM embeddings underperform relative to CLIP and self-supervised vision models, as reported on the MIEB leaderboard [47]. All evaluations are conducted using the official MIEB codebase [76].

- **Visual STS** [74]: It conceptualizes traditional semantic textual similarity (STS) as a vision task by rendering text as images and evaluating the semantic understanding of visual encoders. Similarity scores are computed from the embeddings of image-text pairs and compared against human annotations using Spearman correlation. This task comprises three subcategories: *English* (STS-12∼16), *cross-lingual* (STS-17, with image pairs in different languages, *e.g.,* Arabic–English), and *multilingual* (STS-b, with pairs in the same language, *e.g.,* Italian–Italian). Visual STS naturally assesses a model's interleaved encoding ability to capture semantic meaning from text in image form, with **Spearman correlation** as the primary evaluation metric.

- **Multilingual Image Retrieval**: We utilize the WIT datasets [57] and select its image retrieval subtasks across 11 different languages. This task accesses a model's alignment capability between image and multilingual text embeddings with **nDCG@10** as the main metric.

- **Document Understanding**: We select 7 tasks from the Vidore benchmark [19], which is to retrieve visual documents that contain information to solve the problem in the query. This task assesses a model's ability to handle the complex layouts in visual documents and the interleaved image-text alignment. Here we use **nDCG@5** as the evaluation metric.

- **Image Linear Probing**: We evaluate three linear probing tasks—Stanford Cars [33], BirdSnap [3], and Country211 [50]—which MLLMs struggle the most, as indicated by the MIEB benchmark leaderboard. We follow Xiao et al. [76] to adopt 16-shot linear probing, which closely preserves ranking compared to full-dataset probing, and report **accuracy** as the metric.

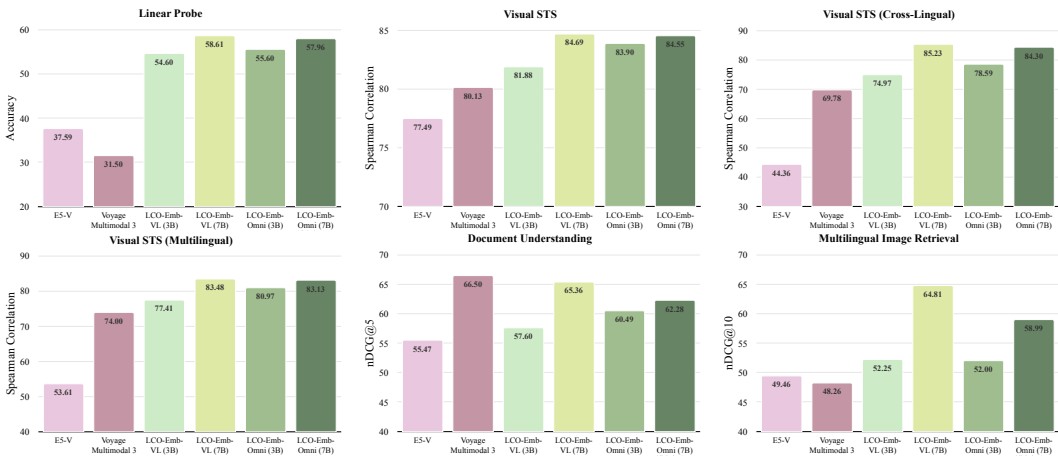

Figure 6: Comparison between the **text-only** variants of LCO-EMB with advanced open-source (E5-V [30]) and proprietary (Voyage Multimodal 3 [65]) embedding models on **MIEB-Sub18**.

Table 3: Comparison of different LoRA ranks and alpha values using Qwen2.5-VL-7B as the backbone. $*r = 256$, $\alpha = 512$ setting experiences unrecoverable loss spikes in training.

| Rank ($r$) | Alpha ($\alpha$) | Comp. | VC-QA | Multiling. Img. Rtr | V-STS (eng.) | V-STS (cross) | V-STS (multi) | Doc. Und. | Linear Probe | Average |
|---|---|---|---|---|---|---|---|---|---|---|
| 8 | 16 | 48.35 | 58.08 | 56.64 | 85.05 | 85.30 | 83.48 | 67.49 | 53.91 | 67.29 |
| 64 | 16 | 55.64 | 60.62 | 55.62 | 84.60 | 85.16 | 83.40 | 65.76 | 52.44 | 67.90 |
| 64 | 128 | 43.40 | 51.86 | 58.93 | 84.98 | 84.44 | 83.39 | 67.66 | 57.24 | 66.49 |
| 256 | 16 | 52.29 | 57.30 | 57.49 | 84.88 | 85.68 | 83.61 | 66.95 | 53.36 | 67.70 |
| 256 | 128 | 43.07 | 57.56 | 56.32 | 85.89 | 84.82 | 83.98 | 67.33 | 55.51 | 66.81 |
| 256 | 512 | 85.52* | 39.24* | 0.70* | 5.90* | 12.90* | 7.80* | 0.90* | 1.50* | 19.31* |

# E   Impact of LoRA Hyperparameters for LCO-EMB

Our approach, LCO-EMB, employs LoRA fine-tuning for lightweight contrastive learning, which aims to refine MLLM representations while minimally perturbing the model's intrinsic knowledge and abilities, thereby effectively preserving its inherent cross-modal alignment capability. We encapsulate this benefit as the "learn less, forget less" characteristic of LoRA. Here, we further analyze the impact of two critical LoRA hyperparameters—rank ($r$) and $\alpha$—on the performance of LCO-EMB.

In LoRA, rank ($r$) and alpha ($\alpha$) jointly control the capacity for new knowledge integration and the extent to which it modulates existing knowledge. The rank defines the dimensionality of the trainable weight matrices used to approximate the original model's weight updates; a higher rank thus increases the capacity for injecting new knowledge. Conversely, alpha scales the contribution of these matrices to the overall model weights, meaning a larger alpha amplifies the extent to which this new knowledge is infused into the model.

Table 3 presents the results of LCO-EMB under different values of rank and alpha. We observe distinct patterns for different task categories, and there doesn't exist a global optimal setting of LoRA hyperparameters. For instance, LoRA hyperparameters bring minimal variations to tasks optimized by the training (*e.g.,* V-STS, whose textual counterpart STS is deemed directly optimized by All-NLI in text embedding literature, is invariant to LoRA hyperparameters). The optimal performance for multilingual retrieval, document understanding, and image linear probe generally occurs when alpha $\alpha$ is scaled up appropriately to rank $r$, such as $r = 8, \alpha = 16$ and $r = 64, \alpha = 128$. However, we notice that for tasks whose capabilities assessed largely differ from those which the training set optimizes, *e.g.,* compositionality and vision-centric QA, a larger alpha $\alpha$ generally brings significant performance degradation, showing the importance of the preservation of the base model's knowledge for generalization to OOD tasks. We observe that with rank 256 and alpha 512, models experience unrecoverable loss spikes in training.

Table 4: Exploring the impact of training dataset utilization and model ensemble on LCO-EMB, where LCO-EMB-Ens denotes the ensemble model produced by applying the model soup [71] technique to LCO-EMB variants fine-tuned on all-NLI and Scale-1M.

| Model | Data Source | Linear Prob. | v-STS (Eng.) | v-STS (cross) | v-STS (multi) | Doc. Und. | Multi. Img. Rtr. | Avg. |
|---|---|---|---|---|---|---|---|---|
| LCO-EMB | all-NLI | 51.86 | **84.69** | **85.23** | **83.48** | **65.36** | 56.37 | 71.17 |
| LCO-EMB | Scale-1M | **58.61** | 81.27 | 81.42 | 78.42 | 62.12 | **64.81** | 71.11 |
| LCO-EMB-Ens | - | 55.69 | 83.79 | 84.88 | 82.82 | 63.15 | 62.67 | **72.17** |

We acknowledge that an optimal rank and alpha likely exist for models of each size, striking a balance between introducing new knowledge and the extent to which it modifies pretrained model weights. We leave a more comprehensive empirical analysis and theoretical study to quantify this relationship for future work.

## F  Impact of Different Training Datasets and Model Merging for LCO-EMB

Recognizing that language models fine-tuned on different datasets often demonstrate distinct strengths, we independently fine-tune LCO-EMB, using Qwen-2.5-VL-Instruct as the backbone model, on all-NLI and Scale-1M via contrastive learning, then assess the performance of each variant in isolation. Subsequently, we investigate the effect of model ensembling by applying the model soup [71] technique, which merges the parameters of multiple separately fine-tuned models by averaging their weights. The results, presented in Table 4, provide the following three key insights:

- **Performance of all-NLI fine-tuned variant.** LCO-EMB trained by all-NLI excels in Visual STS and Document Understanding, indicating that NLI supervision sharpens not only textual similarity perception but also generalizes to improve their ability to preserve vision-text semantic similarity.
- **Performance of Scale-1M fine-tuned variant.** LCO-EMB adapted by Scale-1M leads on Linear Probing and Multilingual Image Retrieval tasks. Since Scale-1M supplies semantically rich descriptions of real-world scenes, LCO-EMB fine-tuned on this corpus appears to emulate image–caption pre-training, thereby activating image representations without explicit visual data.
- **Performance of model ensemble.** The LCO-EMB-Ens, through merging the LCO-EMB variants trained by all-NLI and Scale-1M, achieves the best overall performance, demonstrating that the model ensemble strategy effectively integrates the complementary strengths of each checkpoint.

## G  Preliminaries of Theoretical Analysis

### G.1  Definitions

**Definition 1** (Population and Empirical Risk). Let $\mathcal{D}$ be the data distribution. The **population contrastive risk** for a model $\theta$ is its true expected InfoNCE loss:

$$\mathcal{L}_c^{\text{pop}}(\theta) := \mathbb{E}_{(X,Y) \sim \mathcal{D}} \left[ \mathcal{L}_{\text{InfoNCE}}(X, Y; \theta) \right]. \tag{4}$$

Given a training set $\mathcal{S} = \{(X_i, Y_i)\}_{i=1}^n$ of size $n$, the **empirical contrastive risk** is defined as:

$$\hat{\mathcal{L}}_c^{\text{emp}}(\theta) := \frac{1}{n} \sum_{i=1}^n \mathcal{L}_{\text{InfoNCE}}(X_i, Y_i; \theta). \tag{5}$$

**Definition 2** (Generative Quality of the Prior). Let $P$ be the prior distribution over the parameters of a pre-trained autoregressive generative model. In the common case where $P = \delta_{\theta_0}$ is a point mass of the model parameters at a pretrained checkpoint $\theta_0$, we define its **generative quality** via the mutual information captured by $\theta_0$:

$$I_P(X; Y) := I_{\theta_0}(X; Y). \tag{6}$$

Under the standard approximation that the generative cross-entropy loss $\mathcal{L}_g(P)$ estimates the conditional entropy (ref. Appendix I), we have the following approximation:

$$\mathcal{L}_g(P) \approx H(Y) - I_P(X; Y) \quad \Longrightarrow \quad I_P(X; Y) \approx H(Y) - \mathcal{L}_g(P). \tag{7}$$

Here, $H(Y)$ is the Shannon entropy of the target data distribution, which quantifies the inherent diversity and complexity of the target modality (*e.g.,* text). Thus, for a fixed dataset, a lower generative loss $\mathcal{L}_g(P)$ corresponds to a higher mutual information and therefore a higher generative quality.

### G.2 Justification for Hypothesis 1

A high $I_P(X;Y)$ implies that the representation of the generative model before contrastive finetuning, $f_P(X)$, is already predictive of $Y$. This pre-existing alignment means positive pairs are closer in representation space, enabling contrastive optimization to reach a lower empirical loss. The term $\log N - I_P(X;Y)$ is the theoretical lower bound on InfoNCE loss under ideal conditions. This hypothesis aligns with our empirical findings in Section 5.1, which show that stronger pretrained generative models yield better representations for downstream tasks, and is consistent with recent literature on decoder-based embedding models (e.g., [86]).

## H  Proof of Theorem 1

*Proof.* We begin with the standard PAC-Bayesian generalization bound, which holds with probability at least $1 - \delta$ for any posterior $Q$:

$$\mathbb{E}_{\theta \sim Q}\left[\mathcal{L}_c^{\mathrm{pop}}(\theta)\right] \leq \mathbb{E}_{\theta \sim Q}\left[\hat{\mathcal{L}}_c^{\mathrm{emp}}(\theta)\right] + \sqrt{\frac{\mathrm{KL}(Q\|P) + \log(1/\delta)}{2n}}. \tag{8}$$

According to Hypothesis 1, the empirical risk is bounded as:

$$\mathbb{E}_{\theta \sim Q}\left[\hat{\mathcal{L}}_c^{\mathrm{emp}}(\theta)\right] \leq \log N - I_P(X;Y) + \epsilon_P. \tag{9}$$

Substituting (9) into (8) yields Theorem 1. $\square$

## I  Relationship Between Generative Loss and Conditional Entropy

**Definition 3** (Generative Quality of the Prior). Let $P$ be the prior distribution over the parameters of a pre-trained autoregressive generative model, centered at $\theta_0$. We define its **generative quality** via the mutual information $I_P(X;Y) := I_{\theta_0}(X;Y)$ that its representations capture between the input $X$ and the target output $Y$.

The mutual information is defined as $I_P(X;Y) = H(Y) - H(Y|X)$. While the true conditional entropy $H(Y|X)$ is unknown, it can be estimated by the model's generative cross-entropy loss, $\mathcal{L}_g(P)$. The formal relationship is:

$$\mathcal{L}_g(P) = H(Y|X) + D_{\mathrm{KL}}(p_{\mathrm{data}}(Y|X) \| p_{\theta_0}(Y|X)), \tag{10}$$

where $p_{\theta_0}$ is the model's predictive distribution. For a well-trained MLLM, the goal of minimizing generative loss is to minimize this KL divergence. Thus, for a strong prior, we can use the approximation $H(Y|X) \approx \mathcal{L}_g(P)$. Substituting this into the definition of mutual information yields:

$$I_P(X;Y) \approx H(Y) - \mathcal{L}_g(P). \tag{11}$$

Here, $H(Y)$ is the entropy of the target data, which is constant for a given dataset. Therefore, a **lower generative loss** $\mathcal{L}_g(P)$ directly corresponds to **higher mutual information** and thus a higher generative quality of the prior.

## J  Improving Representation Bounds via Enhancing Generative Capability

To further investigate the hypothesis that enhancing an MLLM's generative ability improves its representations, a key aspect of the Generation-Representation Scaling Law, we introduce a challenging cross-lingual multimodal document retrieval task, **SeaDoc**. This task enables a comprehensive evaluation of MLLM's representational capacity. In this task, an English query is used to retrieve a corresponding multimodal document page in a low-resource target language.

### J.1  Data Curation

SeaDoc is a cross-lingual visual document retrieval benchmark specifically designed for low-resource **S**outh**E**ast **A**sian (SEA) languages. While building upon foundational concepts from existing visual document understanding benchmarks like ViDoRe [19], SeaDoc uniquely challenges MLLMs' visual document understanding capabilities on non-English languages at an unprecedented scale.

Specifically, we utilize Gemini-2.5-Flash [12] to annotate each PDF page by sequentially applying OCR, translating the content into English, and generating an English query answerable exclusively from that specific page. This results in $5,055$ annotated {OCR, English translation, English query} triplets. To construct a high-quality query pool for the retrieval dataset in SeaDoc, we implement a three-stage quality control process:

1. Qwen2.5-7B-Instruct is first used to filter out functional pages (*e.g.,* title pages, author pages, tables of contents), which reduces the dataset to $4,491$ content page annotations.
2. The same model then scores these annotations for *Quality* and *Groundedness* on a 10-point scale. Only questions with a quality score of at least **9** and a groundedness score of **10** are retained. Note that *Quality* measures the informativeness of the content and relevance of the query, and *Groundedness* measures the exclusivity of the answer to the page.
3. Our in-house linguists conduct a final review of the remaining triplets to ensure their quality.

As a result, we derive $1,001$ high-quality queries to be used for retrieval tasks within the $5,055$ page corpus.

For conducting additional OCR-intensive generative training, we construct a training set leveraging images that do not correspond to retrieval test set queries, resulting in 4k seed images. We construct 5 SFT tasks per image: 1) OCR the image. 2) OCR the image, then generate a question from the image. 3) Provide the English translation given the OCR'd text. 4) Provide the English translation directly from the image. 5) Provide the answer to the generated query. Note that compared to the SeaDoc test set, the training set is separately generated and includes an additional "provide answer to the generated question" part in the seed prompt. This process leads us to an around 20k training set to enhance targeted generative capability on low-resource visual documents, which we also explore combining with the PixmoCap dataset (710k) for general capability preservation in the main experiments.

### J.2  Experimental Settings

We use Qwen2.5-VL-3B as the backbone and establish a baseline with lightweight contrastive learning. To assess whether enhanced generative ability benefits embedding quality, we further train a variant with additional generative pre-training before lightweight contrastive learning.

We apply supervised fine-tuning to enhance the model's image-to-text generative capability. This stage utilizes a mixture of image-to-text training data, comprising *OCR data in SEA languages* (derived from the training split of SeaDoc) and *general-domain image captioning data*, *i.e.,* PixmoCaps [13]. The OCR data strengthens its capability in generating SEA languages from visual documents, while the inclusion of general image captioning data helps preserve its semantic alignment between image and text modalities in the general domain.

Given that text in multimodal documents can be small, requiring higher image resolution for MLLMs to accurately read textual content, we

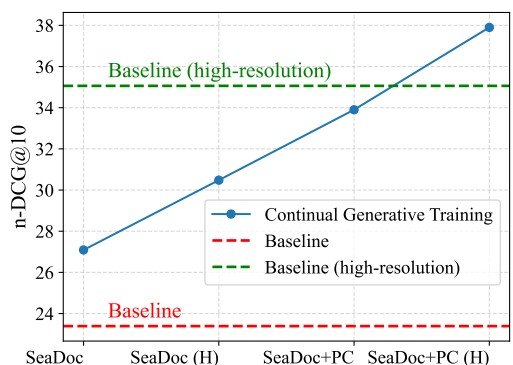

Figure 7: Retrieval performance of Qwen2.5-VL-3B fine-tuned on various continual generative fine-tuning strategies before CL on SeaDoc benchmark, where "PC" represents PixmoCaps and "H" denotes high-resolution. The results suggest that enhancing the generative ability of MLLMs before CL can enhance their embedding capability.

further employ two settings—high- and low-resolution—to assess the impact of image resolution. For the low-resolution setting, we follow standard practice by using a maximum of $262,144$ pixels [87].

For the high-resolution setting, we use a $10\times$ larger maximum of $2,621,440$ pixels. Here, we adopt **nDCG@10** as the primary metric.

## J.3 Experimental Results

Figure 7 summarizes the retrieval performance of the same backbone model fine-tuned using different continual SFT strategies before the same CL tuning process on our SeaDoc benchmark.[5] We draw the following key observations:

(1) When SFT training is conducted exclusively on OCR-intensive data, *i.e.,* SeaDoc-train, at lower resolution, the model experiences a significant capability collapse compared to the baseline (Qwen2.5-VL-3B after lightweight CL). This SFT-induced degradation aligns with observations in recent multimodal reasoning research [7, 15, 28, 36, 66, 81]. Since foundation models have already undergone extensive SFT and RL, continual SFT can lead to overfitting and degrade models' generalization capability.

(2) Training on SeaDoc with higher resolution partly mitigates this collapse. This is because the text in visual documents is typically small; training with higher resolution allows for better grounding of the generated output in the visual text of the source image, as opposed to overfitting to example-level visual cues.

(3) Incorporating PixmoCaps captions into the training set further boosts visual document retrieval performance post-CL. This is because general-domain image captioning data helps preserve the latent image-text alignment learned by MLLMs during pre-training. This preserved alignment is crucial for its effective exploitation by the subsequent text-only contrastive finetuning process.

---

[5]Unless otherwise specified, we evaluate model performance at the maximum resolution used during training.

