# OpenReview forum: "Scaling Language-centric Omnimodal Representation Learning"
_NeurIPS.cc/2025/Conference — NeurIPS 2025 poster_

### Official Review · Reviewer_FpEU · 2025-06-12

**Clarity:** 3
**Significance:** 3
**Originality:** 3
**Rating:** 5
**Confidence:** 3

**Summary:**

This paper empirically demonstrates that latent alignment emerges within MLLM representations, allowing contrasive learning to serve as a lightweight refinement stage. The authors first propose two ways to identify that text-only contrasive learning is beneficial for the multimodal ability of MLLMs, and then propose Language-centric Omnimodal Representation Learning to improve the multimodel performance of MLLMs.

**Questions:**

See above

**Ethical Concerns:**

["NO or VERY MINOR ethics concerns only"]

**Final Justification:**

Considering the rebuttal, i think this paper is novel with sufficient experiemental results. I will maintain my positive rating and lean towards accepting this paper.

**Limitations:**

Yes

**Quality:**

3

**Strengths And Weaknesses:**

Strength:
1. The paper is well-written and easy to follow, with a logical flow and clear explanations of the proposed methodology. The overall readability makes it accessible not only to experts in the field but also to researchers and practitioners who may be less familiar with the specific domain.
2. The experimental results demonstrate that the proposed method performs well across a variety of benchmark datasets. This consistency suggests that the approach is robust and effective in different settings, which enhances its credibility and practical value.
3. The proposed method introduces a novel strategy that can be applied to a wide range of Multi-Modal Large Language Models (MLLMs). Its adaptability across different architectures and tasks indicates that it has the potential to become a valuable tool for future research and real-world applications in multi-modal learning.

Weaknesses:
1. In Section 2.1, the authors mention conducting text-only contrastive learning, but the process is not clearly explained. Specifically, it would be helpful to clarify how the contrastive loss is formulated in the absence of visual inputs, what kind of data is used, and how this component contributes to the overall training objective. Providing more details would improve reproducibility and deepen the reader’s understanding of the method.
2. The experiments appear to rely on LoRA (Low-Rank Adaptation) for training the language decoder. However, it is unclear how the performance would change if the model were trained without LoRA—i.e., through direct fine-tuning of the full decoder. A comparison between these two approaches would help assess whether the use of LoRA affects performance or generalization capabilities.
3. Section 4.2 presumably presents an ablation study or some form of analysis; however, the specific results are not clearly reported in the current version. Including concrete numerical results or visualizations would allow readers to better understand the findings and evaluate the effectiveness of the design choices made by the authors.

Minor:
1. Line 97 has no citations.

---

> ### Author Rebuttal · Authors · 2025-07-31
>
> We appreciate your positive comments and insightful suggestions.
>
> **Details of text-only CL in Section 2.1** The key process is similar to that described in Section 3.1. We first decouple the LLM from the MLLM and train it using a standard contrastive loss (InfoNCE loss), where each positive pair consists of two texts; we then plug the LLM back to the MLLM after training. For the experiments in Section 2.1, we use the all-NLI dataset, in which each positive pair is composed of an anchor and an entailment from a natural language inference dataset; for example, Anchor: "Children smiling and waving at the camera"; Entailment: "There are children present." All other texts within the same batch are treated as in-batch negatives. In contrast, for the main experiment in Section 3.1, we employ an additional dataset we constructed, Scale1M, which incorporates a diverse range of multilingual data sources.
> We appreciate your suggestion and will enrich the descriptions in Section 2.1 accordingly!
>
>
> **Performance of Full-Finetuning vs. LoRA** We provide full-finetuning results as follows, which provides decent performance but lags behind LoRA training, when trained with the same Qwen2.5-VL 3b backbone.
> | Size | Model Type | Training Time | Multilingual Retrieval | Visual STS (En) | Visual STS (Cross) | Visual STS (Multi) | Document | Linear Probe | Average |
> |------|--------------------------------|---------------|------------------------|-----------------|--------------------|--------------------|----------|--------------|---------|
> | 3b | Base Model | - | 31.73 | 73.82 | 59.03 | 68.57 | 28.82 | 46.96 | 51.49 |
> | 3b |Text-only (Full-Finetuning) |~2 hours | _44.61_ | _81.65_ | _68.67_  | _77.75_ | _49.71_ | _50.21_ | _62.10_ |
> | **3b** |**Text-only (Lora)** |**~1 hour** |**51.61** |**81.88** |**74.97** |**78.30** |**57.90** |**53.05** |**66.28** |
>
>
> We hypothesize that full-finetuning introduces more perturbation to the pretrained parameter space, which stores latent cross-modal alignment. In contrast, LoRA beter preserves this alignment through minimal adaptation of model parameters.
>
> **The experiment in Section 4.2**
> The experimental results discussed in Section 4.2 are visualized in Figure 5. After establishing the theoretical relationship between the generative and representational capabilities of MLLMs, we compare the base models' performance on generative benchmarks with their post-contrastive learning performance on embedding benchmarks, showing a positive scaling correlation between generation and representation capability, supporting our theoretical analysis.
>
> **Missing citations** Thanks for spotting the typo. We will add the citation to LLaMA in Line 97 in the revised version.

---

> > ### Comment · Reviewer_FpEU · 2025-08-05
> > **Response**
> >
> > After reading the rebuttal, i will maintain my positive rating and lean towards accepting this paper.

---

> > > ### Author Response · Authors · 2025-08-05
> > >
> > > Thank you again for your supportive comments throughout, and insightful suggestions which helped strengthen the paper!

---

### Official Review · Reviewer_GWW4 · 2025-06-30

**Clarity:** 2
**Significance:** 3
**Originality:** 3
**Rating:** 4
**Confidence:** 3

**Summary:**

The paper investigates why multimodal large language models (MLLMs) fine-tuned with contrastive learning (CL) outperform traditional CLIP-style models. It argues that MLLMs benefit from implicit cross-modal alignment gained during generative pretraining. Text-only CL can then serve as a lightweight refinement step that enhances this latent alignment across modalities, eliminating the need for large-scale multimodal pairings during fine-tuning.

**Questions:**

Please answer the questions mentioned in the Weaknesses section.

**Ethical Concerns:**

["NO or VERY MINOR ethics concerns only"]

**Final Justification:**

Since my concerns have been addressed, I updated my score accordingly.

**Limitations:**

Yes

**Quality:**

3

**Strengths And Weaknesses:**

**Strengths:**

1- The paper addresses an important topic, i.e., understanding why MLLM-based approaches are outperforming CLIP-like models.

2- Providing analysis in Section 2 through the use of geometric metrics, such as anisotropy degree and kernel-level similarity.


**Weaknesses:**

1. The paper claims that "a crucial advantage of MLLM-based approaches stems from implicit cross-modal alignment achieved during generative pretraining." However, the analyses and experiments presented seem to indicate that text-only contrastive learning plays a significant role in boosting alignment between modalities and enhancing the textual and non-textual representations. This appears to be different from the core claim.
2. It’s unclear whether the experiments and analyses support the conclusion that text-only contrastive learning enhances both textual and non-textual representations. From the presented results, improvements seem largely attributable to better textual representations. This may, in part, be due to addressing issues such as modality forgetting in MLLMs. Clarifying whether non-textual features also improve, and providing supporting evidence, would strengthen the paper's claims.

3. Ambiguities in Section 2.1 – Anisotropy Analysis:
- The experimental setup for the anisotropy analysis is not fully detailed. For instance, how many samples were used to compute the anisotropy? Without this, we don’t know that the approximation is fairly accurate for the degree of anisotropy.
What exactly does the layer index in Figure 1 refer to, and why does similarity decrease with increasing layer depth?


- The Qwen2.5-Omni model shows a high degree of anisotropy collapse, yet still performs well on benchmarks. How is this explained?


- One of the datasets used, PixmoCap, includes long textual descriptions. Could this influence similarity measurements (e.g., shared phrases inflating similarity)? And what is the effect on the fine-tuning with contrastive learning?


4. Ambiguities in Section 2.2 – Kernel-level Similarity:
- In Figure 2, the 3B model appears to have more layers than the 7B model. Why is this the case?


- The difference between the original and contrastively fine-tuned models is relatively small in the figure. This raises the question: are improvements mainly due to better textual representations rather than improved multimodal alignment?


Prior work has shown that MLLMs can forget text tasks post-pretraining. Could the improvements observed here simply reflect restored text capabilities, rather than enhancements to multi-modal representations?


5. Figure 4 lacks baseline performance metrics. If Qwen2.5-VL and Qwen2.5-Omni were fine-tuned, it's essential to show their original (pre-fine-tuning) performance for comparison. This would help quantify the effectiveness of the fine-tuning process and the contribution of the proposed method.

**Minor Weaknesses:**

The citation for llama at line 97 is missing

---

> ### Author Rebuttal · Authors · 2025-07-31
>
> Thanks for your detailed comments. We aim to address your concerns as follows:
> ### **1. Clarifying Core Claim**
> We clarify the logic between our experiment findings and the core claim as follows:
>
> Our experiments demonstrate that applying text-only contrastive learning to MLLMs not only improves the isotropy of the embedding space in the text modality, but **also generalizes to enhance isotropy in non-textual modalities—even without finetuning on multi-modal data**. The successful transfer of these improvements across modalities provides empirical evidence that MLLMs inherently maintain geometrically aligned latent spaces among different modalities. We will emphasize this finding in the revised version.
>
> We have provided quantitative results on the degree of anisotropy in Section 2.1. The degree of anisotropy is a standard metric for approximating isotropy, with lower values indicating higher isotropy. In lines 72 to 75, we mentioned that "we examine whether this text-specific fine-tuning effectively mitigates anisotropy in non-textual modalities without requiring explicit multimodal training. Successful transfer of improvements across modalities would provide empirical evidence that MLLMs inherently maintain geometrically aligned latent spaces across different modalities."
>
> ### **2. Whether other modality features improve**
> We provide extensive results on the performance of our method in embedding tasks across different modalities in Figure 4, 5 and Table 1, including both uni-modal tasks (**image-only**: linear probing and visual STS) and cross-modal tasks (**image-text**: document understanding, multilingual image retrieval; **audio-text**: audio retrieval; **video-text**: video retrieval). In the table below, we present the embedding performance of the MLLM before and after text-only contrastive finetuning to enable a direct comparison.
>
> | Model                        | Multilingual Retrieval | Visual STS (En) | Visual STS (Cross) | Visual STS (Multi) | Document | Linear Probe | Average |
> |------------------------------|------------------------|-----------------|--------------------|--------------------|----------|--------------|---------|
> | Qwen-VL-3b (vanilla)         | 31.73                  | 73.82           | 59.03              | 68.57              | 28.82    | 46.96        | 51.49   |
> | **Qwen-VL-3b (after text-only CL)**  | **51.61**                  | **81.88**           | **74.97**              | **78.30**              | **57.90**    | **53.05**        | **66.28**   |
> | Qwen-VL-7b (vanilla)         | 40.31                  | 72.77           | 65.25              | 70.96              | 29.01    | 51.77        | 55.01   |
> | **Qwen-VL-7b (after text-only CL)**  | **56.37**                  | **84.69**           | **85.23**              | **83.48**              | **65.36**   | **51.86**        | **71.17**   |
> | Qwen-Omni-3b (vanilla)       | 42.08                  | 71.98           | 68.35              | 71.01              | 20.33    | 50.27        | 54.00   |
> | **Qwen-Omni-3b (after text-only CL)** |  **51.99**                  | **83.90**           | **78.59**              | **80.97**              | **54.82**    | **55.21**        | **67.58**   |
> | Qwen-Omni-7b (vanilla)       | 42.62                  | 71.58           | 68.28              | 69.90              | 25.36    | 51.71        | 54.91   |
> | **Qwen-Omni-7b (after text-only CL)** | **53.36**                  | **84.55**           | **84.30**              | **83.13**              | **56.42**    | **52.71**        | **69.08**   |
>
> | Model                        | Video Retrieval |
> |------------------------------|-----------------|
> | Qwen-VL-3b (vanilla)         | 47.83           |
> | **Qwen-VL-3b (after text-only CL)**  | **66.77**           |
> | Qwen-VL-7b (vanilla)         | 50.76           |
> | **Qwen-VL-7b (after text-only CL)**  | **66.77**          |
>
> | Model                        | Audio Retrieval |
> |------------------------------|-----------------|
> | Qwen-Omni-3b (vanilla)       | 35.19           |
> | **Qwen-Omni-3b (after text-only CL)** | **47.40**           |
> | Qwen-Omni-7b (vanilla)       | 39.84           |
> | **Qwen-Omni-7b (after text-only CL)** | **49.63**           |
>
> We would like to highlight that, cross-modal retrieval performance would have degraded if non-textual modality spaces do not follow the enhanced textual representation space geometrically.
>
> ### **3. Questions for Section 2.1 Anisotropy Analysis**
> First, we want to highlight that **the lower is the better** for anisotropy estimate value, as we are measuring the expected cosine similarity of **random** examples, instead of paired examples.
>
> ‒ **Setting.** We use 1000 examples for each modality (1000 texts & images, 1000 texts & audios, 1000 texts & videos ) for the 3 settings, respectively, leading to 499,500 cosine similarity values (lower triangular of the similarity matrix) for each setting, and we take the mean for it. This is typically deemed sufficient for approximating anisotropy [1].
>
> ‒ **Why cosine similarity decreases every layer.** Note that as we are measuring cosine similarity expectation of **random**  pairs, and it is a good property for random pairs to have lower similarities (more isotropic, less anisotropic). As the layer progresses, the model contextualizes each input more, leading to more different embeddings for each input, and thus, their cosine similarity with other inputs would become lower.
>
> ‒ **"Qwen-omni is anisotropic but why good on benchmarks?"** After text-only contrastive learning, the expected cosine similarity goes down (less anisotropic) for all modalities, making the embedding space more isotropic. We show in the paper that this leads to state-of-the-art performance on embedding tasks.
>
> ### **4. Questions for Section 2.2 Kernel Analysis**
>
> ‒ **Why Qwen-VL 3b has more layers than Qwen-VL 7b**: According to Table 1 of the Qwen2.5-VL technical report [2], Qwen-VL 3B has more layers in the decoder LLM than Qwen-VL 7B (36 layers vs. 28 layers). The report does not provide an explicit rationale for this architectural decision. We suspect that this configuration may lead to better overall performance.
>
> ‒ **Improvement is small** This section measures cross-modal alignment, instead of the actual benchmark performance. We intend to show that, through text-only contrastive learning, cross-modal alignment actually improves, as opposed to decreasing.
>
> ### **5. Performance gain over base models.**
>
> In point 2 above, we provided extensive results comparing base models.
>
> ### **6. Missing citations**
>
> Thanks for spotting the typo. We will add the citation to LLaMA in Line 97 in the revised version.
>
>
>
> [1] Ethayarajh 2019, How contextual are contextualized word representations? Comparing the geometry of BERT, ELMo, and GPT-2 embeddings
>
> [2] Qwen2.5-VL Technical Report. Qwen Team et al. 2025.

---

> > ### Comment · Reviewer_GWW4 · 2025-08-04
> >
> > I would like to thank the authors for providing answers to my questions.
> >
> > Since my concerns have been addressed, I updated my score accordingly.

---

> > > ### Author Response · Authors · 2025-08-05
> > >
> > > Thanks again for your constructive feedback and reading through our rebuttal! We are pleased to hear that our response has addressed your concerns and appreciate your re-evaluation of our work.

---

### Official Review · Reviewer_oULY · 2025-07-01

**Clarity:** 3
**Significance:** 2
**Originality:** 3
**Rating:** 4
**Confidence:** 4

**Summary:**

The paper introduces a language-centric omni-modal representation learning (LC-OMRL) approach that treats a multimodal LLM’s language decoder as the sole trainable component. After freezing all modality encoders and the projector, the decoder is fine-tuned on text-only contrastive triplets (AllNLI or Scale-1M) using LoRA adapters. The authors argue that generative pre-training of the decoder has already induced a latent cross-modal alignment; the lightweight contrastive stage simply “unlocks” this alignment for similarity matching.

**Questions:**

* Could you add native Qwen-VL/Omni embeddings and other alignment approaches (e.g., CLIP-style contrastive learning) to Figure 4 so that readers can judge absolute and relative gains?
* Have you measured pre- vs post-contrastive cross-modal retrieval (image→text, text→audio, etc.) to directly quantify alignment improvement? If so, please share those numbers; if not, why was this omitted?
* Please provide the t-SNE visualization results for the embedding space.
* What empirical evidence supports choosing LoRA over adding a lightweight projection head or full-decoder tuning?

**Ethical Concerns:**

["NO or VERY MINOR ethics concerns only"]

**Final Justification:**

The authors have addressed my questions during the rebuttal, and I changed my evaluation to positive.

**Limitations:**

Please refer to the weaknesses.

**Paper Formatting Concerns:**

None.

**Quality:**

3

**Strengths And Weaknesses:**

# Strengths
* Updating only the decoder with text contrastive loss is computationally cheap (2 epochs; LoRA rank 64) while yielding SOTA embeddings on a broad benchmark suite.
* Improvements on image, audio, and video retrieval without seeing any non-text signals during fine-tuning support the latent-alignment hypothesis.

# Weaknesses
* The lack of comparisons with native Qwen-VL/Omni embeddings and well-known cross-modal alignment methods (e.g., CLIP-style or cross-modal contrastive learning). Without those references, the superiority of LC-OMRL is less convincing.
* Existing work typically reports cross-modal retrieval (R@k) before/after alignment to quantify improvements. While the paper includes multilingual image retrieval, it does not contrast pre- vs post-training performance, nor does it evaluate retrieval on other modalities (audio, video) explicitly for alignment.
* The “collapsed vs isotropic” embedding spaces are merely schematic; the axes are not defined, and no real data projection (e.g., t-SNE) is given. Readers cannot verify that clusters truly spread out after training.
* The text states two advantages of LoRA, but does not compare with classic alternatives such as adding a shallow projection head or fully fine-tuning the decoder; quantitative ablations are confined to LoRA hyper-parameters.

---

> ### Author Rebuttal · Authors · 2025-07-31
>
> Thank you for your valuable feedback. We address your comments point by point as follows:
> * ### **Comparison with native Qwen-VL/Omni embeddings, and well-known cross-modal alignment methods**:
>
> We present a comparison of results between the native Qwen2.5-VL embeddings, a CLIP-style training baseline, and our text-only CL method on vision-and-language embedding tasks in the Table below. For the CLIP-style baseline, we fine-tune the native Qwen2.5-VL-3b model using 800k image-caption pairs from the PixmoCaps dataset with contrastive learning. The results of our text-only contrastive learning method, which is trained on the all-NLI dataset (270k pairs), are also reported.
> Experiment results show that our text-only method consistently improves the performance of the native Qwen embeddings across all tasks. Moreover, our method outperforms the CLIP-style baseline, while requiring only about 1/50th of the training time.
>
> | Size | Model Type | Training Time | Multilingual Retrieval | Visual STS (En) | Visual STS (Cross) | Visual STS (Multi) | Document | Linear Probe | Average |
> |------|--------------------------------|---------------|------------------------|-----------------|--------------------|--------------------|----------|--------------|---------|
> | 3b | Qwen-VL-3b (vanilla) | - | 31.73 | 73.82 | 59.03 | 68.57 | 28.82 | 46.96 | 51.49 |
> | 3b | Qwen-VL-3b (after CLIP-style CL) | ~50 hours | _25.15_ | _72.51_ | _67.45_ | _65.22_ | _48.91_ | _41.05_ | _53.38_ |
> | **3b** |**Qwen-VL-3b (after text-only CL)** |**~1 hour** |**51.61** |**81.88** |**74.97** |**78.30** |**57.90** |**53.05** |**66.28** |
>
>
>
> The following table also presents the comparison results of Qwen2.5-Omni embedding model before and after finetuned by our text-only CL method on vision-and-language embedding tasks. We will include the results of the CLIP-style baseline in the final version.
> | Model | Multilingual Retrieval | Visual STS (En) | Visual STS (Cross) | Visual STS (Multi) | Document Understanding | Linear Probe | Average |
> |------|------------------------|-----------------|--------------------|--------------------|----------|--------------|---------|
> | Qwen-Omni-3b (vanilla)       | 42.08                  | 71.98           | 68.35              | 71.01              | 20.33    | 50.27        | 54.00   |
> | **Qwen-Omni-3b (after text-only CL)** |  **51.99**                  | **83.90**           | **78.59**              | **80.97**              | **54.82**    | **55.21**        | **67.58**   |
>
> The following tables present the performance of Qwen-VL and Qwen-Omni before and after our text-only CL method on video retrieval and audio retrieval tasks.
> For the results of video retrieval, we report the average of the t2v and v2t recall@5 scores on the msrvtt and ActivityNet datasets. For the results of audio retrieval, we report the average of the t2a and a2t recall@5 scores on the Clotho and AudioCaps datasets.
>
> | Model                        | Video Retrieval |
> |------------------------------|-----------------|
> | Qwen-VL-3b (vanilla)         | 47.83           |
> | **Qwen-VL-3b (after text-only CL)**  | **66.77**           |
> | Qwen-VL-7b (vanilla)         | 50.76           |
> | **Qwen-VL-7b (after text-only CL)**  | **66.77**          |
>
> | Model                        | Audio Retrieval |
> |------------------------------|-----------------|
> | Qwen-Omni-3b (vanilla)       | 35.19           |
> | **Qwen-Omni-3b (after text-only CL)** | **47.40**           |
> | Qwen-Omni-7b (vanilla)       | 39.84           |
> | **Qwen-Omni-7b (after text-only CL)** | **49.63**           |
>
>
> * ### **Visualizing collapsed vs isotropic space**
> We have created visualizations using t-SNE and UMAP data projections. Unfortunately, PDF uploads and link sharing are not permitted in the rebuttal. We will include these visualizations in the camera-ready version given the chance.
>
> We would also like to highlight that we have provided quantitative results on the degree of anisotropy in Section 2.1. Measuring the expected cosine similarity between random samples is a validated method for approximating anisotropy degree, with lower values indicating higher isotropy [1]. Our results demonstrate that our text-only contrastive learning method effectively mitigates anisotropy, thereby enhancing the isotropy of the embedding space.
>
> * ### **LoRA training vs other methods (e.g., Projection Layer or Full-Finetuning)**
> We compare LoRA with a full-finetuning method and a shallow projection baseline that only adds a linear projection layer after the output layer. The results of Qwen2.5-VL-3b are presented in the following table. The shallow projection baseline introduces additional parameters to the model, but does not fully leverage the pretrained latent cross-modal alignment, resulting in only a modest average improvement over native embeddings. While full fine-tuning yields reasonable performance, it still lags significantly behind LoRA training. We posit that this is because the contrastive loss used during fine-tuning differs substantially from the pretraining loss (next word prediction) of MLLM, causing full fine-tuning to introduce large perturbations to the model’s pretrained parameters. These perturbations are more likely to disrupt the model’s pre-established cross-modal alignment.
>
> | Size | Model Type | Multilingual Retrieval | Visual STS (En) | Visual STS (Cross) | Visual STS (Multi) | Document | Linear Probe | Average |
> |------|--------------------------------|---------------|------------------------|-----------------|--------------------|--------------------|----------|--------------|
> | 3b | Vanilla | 31.73 | 73.82 | 59.03 | 68.57 | 28.82 | 46.96 | 51.49 |
> | 3b | Text-only (Linear Projection) |  31.31 | 75.25 | 62.95 | 69.32 | 28.12 | 49.19 | 52.69 |
> | 3b |Text-only (Full-Finetuning) |  _44.61_ | _81.65_ | _68.67_  | _77.75_ | _49.71_ | _50.21_ | _62.10_ |
> | **3b** |**Text-only (Lora)** | **51.61** |**81.88** |**74.97** |**78.30** |**57.90** |**53.05** |**66.28** |
>
>
> Furthermore, full-finetuning is much more compute-intensive for contrastive-learning given the quadratic GPU RAM requirements of contrastive learning loss over batch size. e.g., it would take 16 GPUs of 96G RAM to perform the same full-finetuning experiment on 7B models.
>
> [1] Ethayarajh 2019, How contextual are contextualized word representations? Comparing the geometry of BERT, ELMo, and GPT-2 embeddings

---

> > ### Comment · Area_Chair_gn11 · 2025-08-05
> >
> > Dear reviewer oULY -- can you please read the authors' rebuttal and the other reviewers' and engage with the discussion?
> >
> > Thanks,
> >
> > Area Chair

---

> > ### Comment · Reviewer_oULY · 2025-08-07
> >
> > Thanks for the detailed response. My concerns have been addressed, and I will raise my score.

---

> > > ### Author Response · Authors · 2025-08-07
> > >
> > > Thanks for confirming that the response has addressed your concerns! Your valuable suggestions and re-evaluation are much appreciated.

---

### Official Review · Reviewer_iKcM · 2025-07-05

**Clarity:** 2
**Significance:** 3
**Originality:** 3
**Rating:** 4
**Confidence:** 2

**Summary:**

This paper aims to explore the latent alignment mechanism of MLLM and proposes a GRSL to address this issue. The extensive results demonstrate that the performance surpasses that of existing Open SOTA and API-based methods.

**Questions:**

I have some high-level concerns, and I would be willing to raise my score if the authors can address them.

**Ethical Concerns:**

["NO or VERY MINOR ethics concerns only"]

**Final Justification:**

This paper explores how text-only contrastive learning enhances the quality of omni-modal representations and gives solid experimental results. After reading the rebuttal of other reviewers, I tend to accept.

**Limitations:**

Please refer to the weakness.

**Quality:**

3

**Strengths And Weaknesses:**

Strengths:

1. From the identification of the problem, to the design of the empirical experiments, and finally to the proposal of the solution, the entire process is coherent and flows very well.
2. The theoretical analysis in the Appendix provides a solid foundation for the paper.

Weaknesses:

1. The terms "Open SOTA" and "API" are a bit confusing. Could you replace them with specific works? This would make the comparison clearer and more appropriate.
2. Despite the extensive analysis of the problem, Figure 3 still does not make it clear to me what core issue this paper actually solves.
3. The paper lacks case studies beyond quantitative metrics that would help in understanding the benefits of the proposed method. Could you please add some of these details?

---

> ### Author Rebuttal · Authors · 2025-07-31
>
> Thank you for your constructive comments. We address your high-level concerns as follows:
>
> **Open SOTA and API SOTA**: Thanks for the suggestion. "Open SOTA" and "API SOTA" refer to E5-V and Voyage-multimodal-3, respectively, as cited in line 222 and the caption of Figure 4. To improve clarity, we will also update the legend in Figure 4 to display the actual model names.
>
> **What Figure 3 expresses**: Figure 3 illustrates our proposed method. This figure is intended to outline how text-only contrastive learning enhances the quality of omni-modal representations. Specifically, text-only contrastive learning can effectively mitigate anisotropy in non-textual modalities, even in the absence of explicit multimodal training. We conduct empirical studies in Section 2.1 to demonstrate this effect. As shown in Figure 1, text-only contrastive learning reduces the expected cosine similarity of random text-image, text-audio, and text-video pairs from over 0.8 to around 0.5. This indicates that anisotropy is reduced and the embedding space becomes more discriminative.
>
> **Case studies**:
> We find that conducting text-only contrastive learning significantly improves the model’s fine-grained cross-modal alignment, enabling it to better distinguish between hard-negative texts. For instance, the models perform much better on compositionality evaluation tasks—such as examples from the ARO benchmark—where the goal is to match the correct, fine-grained caption to each image rather than its hard negative.
> As an example, consider a picture of a man sitting to the right of a door, with the following two fine-grained descriptions:
> - "The door is to the left of the shirt" (correct)
> - "The shirt is to the left of the door" (incorrect)
>
> Previously, the model would assign a higher cosine similarity to the incorrect caption. After applying text-only contrastive learning, however, it correctly assigns higher similarity to the accurate caption.
> We appreciate your suggestion and will include such case studies in the paper!

---

> > ### Comment · Reviewer_iKcM · 2025-08-05
> >
> > After reading the rebuttal of other reviewers, i will raise my score.

---

> > > ### Author Response · Authors · 2025-08-05
> > >
> > > Thanks for taking the time to read through the comments! We appreciate your re-evaluation and are available to address any further questions you might have.

---

### Note · Authors · 2025-08-14

We again thank the reviewers for their valuable feedback! We are gratified by the positive comments on the novelty of our language-centric omnimodal representation learning method, which achieves **SOTA performance on key benchmarks** (oULY, FpEU) with **low training compute** (oULY), **advantages over CLIP-based training methods** (GWW4) and **adaptability across architectures** (FpEU). The underlying mechanism is supported by our **cross-modal analyses** (GWW4), which show that latent alignment in pretrained MLLMs is key to our text-only training's generalization across modalities (oULY). We are also happy that reviewers praised the paper's **clear structure** (iKcM), **accessibility for both experts and non-experts** (FpEU), and **the value of our theoretical analysis** of the generation-representation scaling law (iKcM).

We appreciate the active engagement from reviewers and AC during the rebuttal, where we presented the following additional results and clarifications:

- **LoRA vs. Alternative Training Methods** (full-finetuning and linear projection), showing the necessity of using LoRA to preserve the latent cross-modal alignment in MLLMs. We also compared with CLIP-style image-text contrastive learning, showing the advantages of our text-only training.

- **Post-contrastive learning vs. vanilla models' performance:** We provided full results across a suite of unimodal (image-only) and cross-modal (image-text, audio-text, video-text) tasks, confirming that text-only training generalizes to improve representation quality for all modalities and their cross-modal alignment.

We also provided key clarifications regarding our theoretical analyses, including:

- **The logic** why anisotropy analysis (2.1) and kernel similarity analysis (2.2) support the main claim and motivate the training framework: applying text-only contrastive learning improves embedding **isotropy across all modalities (2.1) and cross-modal alignment (2.2), without any multi-modal finetuning**. This successful transfer provides strong empirical evidence that pretrained MLLMs exhibit latent cross-modal alignment, the key mechanism our method leverages.

- **Minor Clarifications:** We clarified the message of Figure 3; and confirmed that Figure 5 displays results supporting our theoretical scaling law analysis (4.2).

We are happy that these clarifications and results have addressed the reviewers' concerns and sincerely hope our work will provide valuable insights to the community!

---

### Decision · Program_Chairs · 2025-09-17

**Decision:**

Accept (poster)

**Comment:**

The submission addresses how multimodal large language models that were fine-tuned with contrastive learning perform and argues that the MLLMs acquire an implicit alignment across modalities during generative pre-training. They use this insight to propose GRSL. The experimental analyses and comparisons demonstrate the benefit.

All reviewers acknowledge the merit of the submission and recommend acceptance, though mainly with a borderline stance. The AC agrees and recommends acceptance.